# The Anticancer Effect of a Novel Quinoline Derivative 91b1 through Downregulation of *Lumican*

**DOI:** 10.3390/ijms232113181

**Published:** 2022-10-29

**Authors:** Yuanyuan Zhou, Zhongguo Zhou, Dessy Chan, Po yee Chung, Yongqi Wang, Albert Sun chi Chan, Simon Law, Kim hung Lam, Johnny Cheuk On Tang

**Affiliations:** 1School of Biomedical Engineering, Sun Yat-sen University, Guangzhou 510006, China; 2State Key Laboratory of Chemical Biology and Drug Discovery, Lo Ka Chung Centre for Natural Anticancer Drug, Development, Department of Applied Biology and Chemical Technology, The Hong Kong Polytechnic University, Hong Kong, China; 3School of Chemistry and Molecular Biosciences, The University of Queensland, Brisbane, QLD 4032, Australia; 4Department of Biosystems Science and Eng, Eidgenössische Technische Hochschule (ETH) Zürich, 4058 Basel, Switzerland; 5School of Pharmaceutical Sciences, Sun Yat-sen University, Guangzhou 510006, China; 6Department of Surgery, Li Ka Shing Faculty of Medicine, The University of Hong Kong, Hong Kong, China

**Keywords:** quinoline, anticancer, *Lumican*, microarray

## Abstract

Quinoline derivatives have been reported to possess a wide range of pharmaceutical activities. Our group previously synthesized a series of quinoline compounds, in which compound 91b1 showed a significant anticancer effect. The purpose of this study was to evaluate the anticancer activity of compound 91b1 in vitro and in vivo, and screen out its regulated target. A series of cancer cell lines and nontumor cell lines were treated with compound 91b1 by MTS cytotoxicity assay and cell-cycle assay. In vivo anticancer activity was evaluated by a xenografted model on nude mice. Target prediction of 91b1 was assessed by microarray assay and confirmed by pancancer analysis. Relative expression of the target gene *Lumican* was measured by qRT-PCR. 91b1 significantly reduced tumor size in the nude mice xenograft model. *Lumican* was downregulated after 91b1 treatment. *Lumican* was proven to increase tumorigenesis in vivo, as well as cancer cell migration, invasion, and proliferation in vitro. The results of this study suggest that the anticancer activity of compound 91b1 probably works through downregulating the gene *Lumican.*

## 1. Introduction

Cancer continues to be a major public health issue globally. Researchers have estimated that there were around 19.3 million new cases of cancer and 10.0 million cancer deaths worldwide in 2020 [1]. Oncogenes, tumor-suppressor genes, and other homeostatic genes are usually responsible for tumorigenesis. Multiple mutations of different genes affect cellular pathways that they control, which also contributes to the cause of cancers [2]. Gene expression, morphology, unregulated proliferation, escape from programmed cell death, and ability to invade distant tissues allow cancer cells to form an aggressive tumor population with metastasis [3]. Thus, the development of anticancer drugs with high efficacy and low toxicity remains a great challenge. Exploring novel oncogenes as a therapeutic target is a promising strategy for precision treatment and individualized treatment.

Quinolines are one of the most important classes of heterocyclic alkaloids, which have been widely reported to possess a broad range of pharmaceutical activities [4,5]. Quinoline-bearing structures exist in many biologically active drugs, including Quinine, Quinodine, Mefloquine, and Chloroquine [6,7,8]. Quinoline and its heterocyclic derivatives have been tested by various groups and proved to have many biological activities, including antimalarial, antibacterial, antiviral, and anti-inflammatory activity [9,10,11]. Quinoline derivatives have also been extensively studied as potential antitumor agents [12,13]. With the development of cytobiology and molecular biology, the essential principles of tumorigenesis, invasion, migration, and metastasis induced by quinoline derivatives have been further explained. Antitumor mechanisms of quinoline derivatives include alkylating DNA [14], inhibiting c-Met kinase [15], epidermal growth factor receptor (EGFR) [16], and vascular endothelial growth factor (VEGF) [16,17,18]. Some were also proven to inhibit P-glycoprotein [19]. The anticancer activity of quinoline derivates has been widely recognized and investigated. Our group synthesized a series of quinoline derivatives and evaluated their anticancer effect. A total of 27 compounds were examined for anticancer activities against cancer cell lines of hepatocellular carcinoma (Hep3B), lung carcinoma (A549), and esophageal squamous cell carcinoma (HKESC-1, HKESC-4, and KYSE150) [20]. Quinoline compound 83b inhibited cancer growth in esophageal squamous cell carcinoma by downregulating COX-2 and PGE2 [21]. The cytotoxic potential of six quinoline derivatives were examined both in vitro and in vivo [22]. 2-Formyl-8-hydroxy-qinolinium chloride was prepared and its anticancer activity was evaluated both in vitro and in vivo [23]. The relationship between structure and activity was also researched, and the presence of a bromine atom and hydroxy group are likely the favorable structural components for molecules to exert a strong anticancer effect [22,24]. The identification of the target of studied compounds is an important step in advancing the field of drug development.

LUMICAN belongs to Class II of the small leucine-rich proteoglycan family (SLRPs), which are among the important noncollagenous extracellular matrix (ECM) proteins [25]. *Lumican* is localized at chromosome 12q21.3-q22 and encodes a 338-residue protein [26], and is expressed in many tissues, including the skin, arteries, lungs, vertebral discs, kidneys, bone, aorta, and articular cartilage [27]. Recent studies have shown that LUMICAN is a key regulator of collagen fibrillogenesis and participates in the maintenance of tissue homeostasis, as well as modulating cellular functions, including cell proliferation, migration, and differentiation [28]. *Lumican* expression is associated with various cancer types, resulting in either protumorigenic or antitumorigenic effects [29]. In breast cancer, LUMICAN protein transforms mesenchymal cells into epithelial-like cells to reverse metastases [30]. In lung cancer, *Lumican* is highly expressed in osteotropic lung cancer cells, with an enhanced capacity of bone metastasis. Downregulation of *Lumican* suppresses cancer cell migration and invasion in vitro [31]. In gastric cancer, *Lumican* expression in tumor tissues is significantly higher than that in adjacent nontumor tissues. A high expression of *Lumican* is an important risk factor for poor survival [32].

In the present study, one of the most promising quinoline compounds, 91b1, synthesized by our group, was evaluated in vitro and in vivo to identify its anticancer activity. To further explain the mechanisms of the anticancer behavior of compound 91b1, the most associated gene, *Lumican,* was screened and studied. The overall results provided a promising candidate for anticancer therapy and evidence of a cancer-related gene, which may suggest a target for further anticancer drug development.

## 2. Results

### 2.1. Compound 91b1 Showed Anticancer Effect In Vitro

The cytotoxicity of compound 91b1 on four cancer cell lines (A549, AGS, KYSE150, and KYSE450) and one nontumor cell line (NE3) was examined by MTS assay. As shown in Figure 1a–e, Compound 91b1 exhibited a comparable inhibitory effect on all tested cell lines with CDDP in a dose-dependent manner. The cytotoxic effect was determined by the MTS_50_ value, which showed a 50% reduction in MTS signal compared with the vehicle control. The MTS_50_ values of compound 91b1 and CDDP on cells are summarized in Table 1. The MTS_50_ values of compound 91b1 were 15.38 μg/mL, 4.28 μg/mL, 4.17 μg/mL, and 1.83 μg/mL for the A549, AGS, KYSE150, and KYSE450 cell lines, respectively. The MTS_50_ values of CDDP were 6.23 μg/mL, 13.00 μg/mL, 13.2 μg/mL, and 6.83 μg/mL for the A549, AGS, KYSE150, and KYSE450 cell lines, respectively. The MTS_50_ values of compound 91b1 were lower than those of CDDP in the AGS, KYSE150, and KYSE450 cell lines, implying that compound 91b1 had a stronger anticancer effect than CDDP in these three cancer cell lines. For the nontumor cell line NE3, the MTS_50_ values of compound 91b1 and CDDP were 2.17 μg/mL and 1.19 μg/mL, respectively, indicating that compound 91b1 may be less toxic than CDDP to nontumor cells.

To study the effect of compound 91b1 on cell proliferation, an MTS cell proliferation assay was performed on A549, AGS, KYSE150, and KYSE450 cell lines. The results are shown in Figure 1f–i. Cell lines treated with compound 91b1 showed an obvious reduction in proliferation rate compared with the vehicle control group on A549 cells after 48 h of incubation, AGS cells after 72 h of incubation, KYSE150 cells after 24 h of incubation, and KYSE450 cells after 48 h of incubation, respectively, indicating that compound 91b1 inhibited cancer cell growth at an incubation time of 48 h, except AGS cells (cells were significantly inhibited after 72 h), but more strongly inhibited KYSE150 cell proliferation (cells were significantly inhibited after 24 h of incubation).

To explain the effect of low proliferation rate in cancer cells after compound 91b1 treatment, cell-cycle analysis was performed to reveal the cell-cycle changes of A549 and KYSE450 with 91b1 treatment. As shown in Figure 1j,k. The percentage of distribution of the G0/G1 phase in A549 cells was increased dose-dependently after the compound 91b1 treatment. The percentage of distribution of the G0/G1 phase in KYSE450 cells showed an increasing trend after treatment with compound 91b1; however, no significant difference was observed. It is suggested that compound 91b1 induced A549 cancer cells and KYSE450 cancer cells to be arrested at the G0/G1 phase and finally resulted in the inhibition of cancer cell growth.

### 2.2. In Vivo Antitumor Effect of Compound 91b1 on Nude Mice Xenograft with KYSE450

To further evaluate the antitumor effect of compound 91b1, athymic nude mice xenografted with KYSE450 cells were used in this project. Figure 2 summarizes the change of relative tumor volume of nude mice xenografted with KYSE150 cells with the treatment of compound 91b1 via images of nude mice in the treated group and control group on the initial day and 25th day, respectively. Compound 91b1 significantly inhibited tumor growth in mice at the dose of 50 mg/kg compared to vehicle control. The tumor volume of the nude mice in the vehicle control group increased gradually every day and reached about five times the size of the tumor on the initial day. The average tumor volume of the nude mice with the administration of 50 mg/kg/day compound 91b1 was just two times the volume of the tumor on the initial day. Additionally, in the compound 91b1-treated group, one of the tumors remained at the initial size, and one was totally invisible at the 13th day, without relapse.

### 2.3. Lumican Plays a Critical Role in Several Types of Cancers and Can Be Downregulated by Compound 91b1

cDNA microarray analysis was performed to study the changes of gene expression caused by compound 91b1 in cancer cells. A heatmap (Figure 3a) and violin plot (Figure 3b) showed the different expression profile of KYSE150 cells treated with compound 91b1 compared with a blank control. A scatter plot of different expressed genes (Figure 3c) exhibited upregulated genes as red dots and downregulated genes as green dots. Collectively, there were 31,520 upregulated genes and 13,180 downregulated genes identified in total. Gene expression changes was evaluated by signal-intensity fold change. Figure 3d summarizes the five most downregulated genes and the five most upregulated genes. *Lumican* was downregulated to 48.34% compared with the control group, and was studied in this project. The relative expression of *Lumican* was examined by quantitative real-time PCR on KYSE150 cells after treatment with compound 91b1. As shown in Figure 3e, the *Lumican* mRNA expression level of KYSE150 cells decreased in a dose-dependent manner with increasing concentrations of compound 91b1 (5 μg/mL, 9.5 μg/mL, 20 μg/mL, and 50 μg/mL), and additionally showed significance when the concentration of compound 91b1 reached 20 μg/mL. According to the pancancer analysis in Figure 3f, *Lumican* was differently expressed in several kinds of cancers, suggesting the critical roles of *Lumican* in tumorigenesis and tumor development.

### 2.4. Overexpression of Lumican Promoted Tumorigenesis

To verify the microarray results, the relative expression of *Lumican* in tumor tissues was compared with the adjacent normal tissue after being normalized by the expression of *β-actin* by ΔCt (ΔCt = Cq of *Lumican*–Cq of *β-actin*). The scatter plot is shown in Figure 4a, indicating that the relative expression level of *Lumican* of the tumor sample group was in general higher than in the nontumor sample group. Relative expression levels of *Lumican* in seven cancer cell lines were compared with the nontumor cell line (NE3) after they were normalized by the expression of *β-actin* and calculated by the 2^−ΔΔCt^ method by qPCR. As shown in Figure 4b, five out of seven (71.4%) cancer cell lines—KYSE30, KYSE70, KYSE150, KYSE510, and SLMT1—showed a higher relative expression level of *Lumican* than NE3; one cancer cell line, HKESC3 [34], showed a lower relative expression level of *Lumican* than NE3; while one cancer cell line, KYSE450, showed no significant difference of relative *Lumican* expression compared with NE3. The overall results suggested that *Lumican* is usually overexpressed in tumor tissue or cancer cells. To identify the function of the *Lumican* gene in tumorigenesis and development, NIH3T3 parental, NIH 3T3/Lum, or NIH 3T3/Mock cells were subcutaneously injected into the flanks of female Balb/c athymic nude mice. After a 14-day period, the possible formation of subcutaneous tumors was observed. Figure 4c shows images of each animal in the NIH 3T3 parental group, NIH 3T3/Mock group, and NIH 3T3/Lum group on day 0, day 7, and day 14.

### 2.5. Lumican Induces Cancer Cells Migration, Invasion, and Proliferation

To further study the function of *Lumican*, KYSE150 cells were cocultured with recLumican to investigate the effect of *Lumican* on wound healing and transwell mitrigel invasion. As shown in Figure 5a, after 24 h incubation with recLumican, there were more cells migrated into the scratched area than the control group in a dose-dependent manner. Figure 5b shows images of crystal violet-stained cells, which were transferred through membranes after coculturing with recLumican after 24 h. The average invaded cell numbers were summarized in Figure 5c. The invaded cell number of KYSE150 cocultured with recLumican was increased with the increasing concentration of recLumican compared with the control group. To further study the functional roles of *Lumican* on cancer cell growth, a cell proliferation assay was performed by MTS on A549, AGS, KYSE150, and KYSE450 cells cocultured with or without recLumican at the concentration of 250 ng/mL (based on the results of the cell invasion assay). According to the results shown in Figure 5d–g, the cell lines cocultured with 250 ng/mL recLumican showed an increase in the proliferation rate compared with the control group for A549, AGS, and KYSE150, indicating that *Lumican* can promote the proliferation of cancer cells.

## 3. Discussion

Quinoline compounds isolated from natural sources have been reported to have great potential in pharmaceutical applications [5]. Based on quinoline’s structure, a series of compounds have been synthesized by our group [22,35]. Chemically modified natural compounds possess high biological activity with low toxicity. One of the quinoline compounds we synthesized, 91b1, was studied in this project.

The cytotoxic effect of compound 91b1 on cancer cells compared with noncancer cells was evaluated by MTS cytotoxicity assay [36]. CDDP (cisplatin) and doxorubicin (DOX) are well-known chemotherapeutic drugs to treat NSCLC (nonsmall cell lung cancer), ESCC (esophageal squamous cell carcinoma), and gastrointestinal cancer [37,38,39,40,41,42], which were applied as the positive control to assess the anticancer potential of compound 91b1. The MTS_50_ values of compound 91b1 were lower than those of CDDP in the AGS, KYSE150, and KYSE450 cell lines, implying that compound 91b1 showed stronger anticancer effects than CDDP in these three cancer cell lines. In nontumor cell line NE3, the MTS_50_ value of compound 91b1 (2.17 μg/mL) was higher than CDDP (1.19 μg/mL), indicating that compound 91b1 may be less toxic than CDDP to nontumor cells. Hence, compound 91b1 exhibited a good potential as an anticancer agent, with higher anticancer activity and lower toxicity compared with the first-line anticancer drug CDDP against cancer cells. Furthermore, in vivo tests showed that compound 91b1 significantly suppressed the development of tumors in animals.

The G0/G1 phase of the A549 cell and KYSE450 cell populations were increased, along with the concentrations of compound 91b1. It is suggested that compound 91b1 may induce cancer cells accumulated at the G0/G1 phase and cannot complete the normal cell cycle as usual, which finally results in the inhibition of cancer cell growth. There are also other anticancer agents reported to arrest G0/G1 phase. Peiminine significantly inhibited the proliferation and colony formation of Glioblastoma multiforme by arresting cell-cycle arrest at the G0/G1 phase [43]. Moreover, Casticin induces G0/G1 arrest and apoptosis in gallbladder cancer [44]. These suggest the anticancer potential of compound 91b1 to treat other types of cancers.

*Lumican* was predicted as a possible target of compound 91b1 from cDNA microarray analysis. The protumorigenic effects or antitumorigenic effects of *Lumican* differ in different types of cancers. The mechanisms of *Lumican* have been widely researched recently. *Lumican* was found to be overexpressed in bladder cancer tissues, and the depletion of *Lumican* inhibited bladder cancer cell proliferation and migration by suppressing MAPK signaling [45]. In liver cancer, silencing *Lumican* resulted in decreased cancer cell migration by inhibiting ERK1/JMK signaling [46]. On the contrary, the antitumorigenic effects of *Lumican* usually involve cell–cell communication and epithelial-to-mesenchymal transition [47,48]. In this study, we mainly discuss the protumorigenic effects of *Lumican*. According to our results, *Lumican* was overexpressed in both esophageal patients’ tumor tissue and cancer cell lines (including lung cancer cells, esophageal squamous cell carcinoma, and gastric cancer cells), and promoted cancer cell migration, invasion, and proliferation, which was consistent with previous reports, and offered more evidence of the protumorigenic effect of *Lumican* in these cancers.

Compound 91b1 dose-dependently downregulated the relative expression of *Lumican* in KYSE150 cells, suggesting that quinoline compound 91b1 probably induces an anticancer effect by downregulating the expression of *Lumican,* modulating its upstream or downstream signaling pathway in cancer cells. Bio-Plex Pro Cell Signaling Assay was performed to analyze the involved signaling pathways for treatment with compound 91b1 on KYSE150 cells, to explain the mechanisms of the downregulation effect of *Lumican*. Phosphorylated analytes (AKT (Ser473), ATF-2 (Thr71), MEK1 (Ser217/Ser221), ErK1/2 (Thr202/Tyr204,Thr185/Tyr187), p38 MAPK (Thr180/Tyr182), HSP27 (Ser78), p53 (Ser15), JNK (Thr183/Tyr185), p90 RSK (Ser380), and Stat 3 (Ser727)) from cell lysates, treated with gradually increased concentrations of compound 91b1 (5, 9.5, and 20 μg/mL) or vehicle control, were detected by the Bio-Rad Bio-Plex 200 Suspension Array System. Appendix A summarizes the significant modulated pathways. The reported pathways involved in *Lumican,* ERK1/2, and MAPK will be studied further in the future.

Overall, this study comprehensively evaluates the effects of compound 91b1 and the functions of *Lumican*. The findings of this study provide more proof of the protumorigenic effect of *Lumican* and the potential of quinoline compounds as antitumor agents.

## 4. Materials and Methods

### 4.1. Reagents and Materials

Cell culture medium RPMI-1640, F-12, DMEM (Dulbecco’s modified Eagle’s medium), MEMα (minimum essential medium α), KSFM (keratinocyte serum-free medium), FBS (fetal bovine serum), penicillin, and streptomycin were purchased from Life Technologies (Carlsbad, CA, USA).

### 4.2. Synthesis of Compound 91b1 with ^1^H-NMR Examination

The chiral 5,7-dibromo-2-methyl-1,2,3,4-tetrahydroquinolin-8-ol (R/S 91b) was prepared by asymmetric hydrogenation reaction of 5,7-dibromo-2-methylquinolin-8-ol. 5,7-dibromo-2-methylquinolin-8-ol was synthesized by commercially available 2-methyl-8-quinolinol (1.6 g, 10 mmol), which was dissolved in 150 mL MeOH with dropwise addition of Br_2_ (1 mL) [35]. Dr. Penny Chan from our research group synthesized compound 91b and demonstrated that the antitumor effect of compound (R/S) 91b showed a promising MTS_50_ compared with Cisplatin, but the R enantiomer of 91b (91b1) exhibited better antitumor activity than the S enantiomer of 91b (91b2) on most of the tested cancer lines (Hep3B, HKESC-1, HKESC-4, and KYSE150 cell lines). According to Dr. Penny Chan’s work, we inferred that the chemical structure of 91b1 was more favorable for the cancer cell membrane and could kill the cancer cells more effectively than 91b2. Thus, compound 91b1 was further studied in this project. Compound 91b1 was used in this project to examine in vitro and in vivo anticancer effects. Compound 91b1 was completely dissolved in dimethyl sulfoxide (DMSO). The structure of compound 91b1 was examined by ^1^H-NMR. Figure 6 shows the structure and the ^1^H-NMR spectrum of compound 91b1. The LC/MS report was summarized in Appendix A.

### 4.3. Cell Lines and Cell Culture

A total of 10 cell lines were examined in this study. Six esophageal squamous cell carcinoma (ESCC) cell lines—KYSE30, KYSE70, KYSE150, KYSE450, KYSE510 [49], and HKESC3 [34]—were purchased from DSMZ (Deutsche Sammlung van Mikroorganismen und Zellkulturen, Braunschweig, Germany). One ESCC cell line, SLMT-1, was kindly provided by Professor Gopesh Srivastava of the Department of Pathology, the University of Hong Kong. A lung cancer cell line, A549, and a gastric adenocarcinoma cell line, AGS, were purchased from ATCC (American Type Culture Collection, Manassas, VA, USA). A nontumor esophageal epithelial cell line, NE3 [50], was kindly provided by Professor George S.W. Tsao from the Department of Anatomy of The University of Hong Kong. The culture medium for KYSE30, KYSE150, KYSE450, and KYSE510 was 45% RPMI with 45% F-12 and 10% FBS; that for KYSE70 was 90% RPMI with 10% FBS; that for SLMT-1 and HKESC3 was 90% MEMα with 10% FBS; that for A549 and AGS was 90% DMEM with 10% FBS; and that for NE3 was KSFM with complementary supplements. All media were supplemented with 100 units/mL penicillin G and 100 μg/mL streptomycin, and all cell lines were maintained in a humidified atmosphere of 95% air and 5% CO_2_ at 37 °C. The cultures were passaged at preconfluent densities of about 80% using a solution of 0.25% trypsin (Invitrogen, Waltham, MA, USA). Cells were washed briefly with phosphate-buffered saline (PBS), treated with 0.25% trypsin, and harvested by centrifugation for subculturing.

For the study of *Lumican* tumorigenesis function, NIH-3T3/Lum and NIH-3T3/Mock cells were established from mouse embryo embryonic fibroblast cell line NIH 3T3 cells, which were transfected with *Lumican* expression vector (Human LUM ORF mammalian expression plasmid, C-Myc tag, Sino Bioglogical Inc., Beijing, China.) or mock vector (pCMV/hygro-Negative Control Vector, Myc-tagged, Sino Biological Inc., Beijing, China) as the negative control. NIH-3T3/Lum and NIH-3T3/Mock cells were maintained in DMEM medium supplemented with 10% FBS, 100 μg/mL penicillin, and 400 μg/mL hygromycin (Invitrogen, Waltham, MA, USA) at 37 ℃ in a humidified incubator with 5% CO_2_. Trypsinization was performed when the density of cells reached 80% confluence.

### 4.4. Balb/c Nude Mice

Female Balb/c-nu mice, each weighing 18 g, were purchased from Beijing Charles River Laboratories. The animal approval code was 440072000011798 and the certificate number was SCXK (Beijing) 2012-0001.

The animals were kept in the SPF-grade animal laboratory, which conformed to the SPF grade requirement of an animal testing facility, where temperature was within the range of 22 °C (±2 °C), humidity was within the range of 30~70%, the diurnal lighting and darkness cycle was 12 h, and the number of air changes per hour was within the range of 10–20 times. An individually ventilated cage (IVC) system was applied to culture nude athymic mice. The approval No. of the SPF animal laboratory was SYXK (Guangdong) 2005-0062. The mice chow was SPF-grade full pellets for mouse, which was bought from Guangdong Medicinal Laboratory Animal Center.

The nutritional values and the sanitation condition were confirmed to meet the SPF-grade requirement for animal testing. Antiseptic water was given ad libitum. All animals were quarantined for at least 7 days in a germ-free environment with a 12 h diurnal lighting and darkness cycle to confirm they were in healthy condition for experiments. All animal experiments in this project were conducted following the Cap 340 Animal License from Department of Health (HKSAR Government).

### 4.5. Patient Specimens

Twenty archival esophageal squamous cell carcinoma (ESCC) paired patient specimens (nontumor and tumor) were used for the study of *Lumican* expression. The ESCC tumor specimens were collected from the Department of Surgery, Queen Mary Hospital, Hong Kong, during the period of 1990–2001, after ESCC patients had undergone esophagectomy. Their corresponding nontumor epithelial tissue specimens were collected for comparison, located at least 10 cm away from the tumor.

### 4.6. Cytotoxicity Assay of Compound 91b1

3-(4,5-Dimethylthiazol-2-yl)-5-(3-carboxymethoxyphenyl)-2-(4-sulfophenyl)-2H-tetrazolium (MTS) assay was performed to evaluate the cytotoxic effect of quinoline compounds 91b1 and positive control cisplatin on selected cell lines (cancer cell lines and immortalized nontumor cell lines) using CellTiter96 AQueous One Solution Cell Proliferation (Promega, Madison, WI, USA), following the manual instructions.

Briefly, about 5 × 10^3^ cells were seeded into each well of a flat-bottom 96-well cell culture plate in 100 μL recommended culture medium and were allowed to grow for 24 h at 37 °C with 5% CO_2_. After 24 h of incubation, the old culture medium was replaced by fresh medium with treatment. The concentrations were gradually increased for compound 91b1 or cisplatin, from 0 μg/mL to 50 μg/mL (0, 1.562, 3.125, 6.250, 12.500, 25.000, and 50.000 μg/mL), and 0.1% DMSO was added to the medium as vehicle control, n = 4. The seeded 96-well plates were then incubated for 48 h at 37 °C with 5% CO_2_. The results were recorded by a microplate reader (Bio-RAD, Ultrmark, Microplate Imaging System, Hercules, CA, USA) to measure absorbance at 492 nm to determine the cell viability. The control value corresponding to untreated cells was taken as 100% and the viability of treated samples was expressed as a percentage of the control. Finally, the curves of cell viability against compound concentrations were plotted and the MTS_50_ (concentration of tested compounds that had 50% inhibition against MTS activity) of the tested compounds were determined.

### 4.7. Cell Proliferation Assay by MTS

MTS assay was performed to analyze the effect on cell proliferation of adding the compounds 91b1, doxorubicin, or human *Lumican* protein (recLumican). The tumor cell lines A549, AGS, KYSE150, and KYSE450 were tested with the compound 91b1 or recLumican for the cell proliferating effect.

The tested cells were harvested by trypsinization and the cell number was counted by hemocytometer under a microscope. Approximately 5000 cells were plated in a flat-bottom 96-well plate in 100 μL of respective culture medium in each well. After incubation at 37 °C with 5% CO_2_ overnight, culture medium was replaced by 200 μL of 10 μg/mL compound 91b1, or 250 ng/mL recLumican of proper culture medium as a test group, and 0.1% DMSO culture medium as the vehicle control group. MTS reagent was used for the quantification of cell viability to indicate cell proliferation. MTS working solution was prepared by diluting five times with autoclaved PBS before use (MTS/PBS (*v*/*v*) = 1:4). A total of 100 μL of the MTS working solution was added to each well after removal of the culture medium at 0 h, 6 h, 24 h, 48 h, and 72 h, respectively, without disturbing the attached cells, and then incubated at 37 ℃ with 5% CO_2_ for a period of time, depending on the cell type. The cell viability was then determined by measuring the absorbance of the well at 492 nm using a microplate reader (Bio-RAD, Ultrmark, Microplate Imaging System, Hercules, CA, USA). Relative growth (compared with the cell viability at 0 h) of each cell line was calculated by $\frac{{\left[A\right]}_{T}}{{\left[A\right]}_{T0}}$, where [A]_T_ is the absorbance at different time points and [A]_T0_ is the absorbance at 0 h. This assay was performed in triplicate.

### 4.8. Cell Cycle Analysis

Approximately 8 × 10^5^ cells were seeded into each well of a 6-well plate. After 24 h incubation at 37 °C with 5% CO_2_, culture medium was replaced by fresh medium with compound 91b1 at a gradually increasing concentrations of 5, 10, 20, and 50 μg/mL, doxorubicin at 0.5 μg/mL as positive control, or 0.1% DMSO culture medium as vehicle control (n = 3). After 24 h treatment at 37 °C with 5% CO_2_, cells were harvested by trypsinization to obtain cell pellets to be fixed with 70% ethanol at 4 °C overnight. On the next day, the fixed cells were digested by PI/RNase Staining Buffer (BD Biosciences, San Jose, CA, USA) for 15 min incubation at room temperature in darkness. The samples were analyzed by BD FACSCalibur Flow Cytometer (BD Biosciences, San Jose, CA, USA).

### 4.9. cDNA Microarray Analysis

The cDNA microarray analysis and associated quality control were performed using Human Genome U133 Plus 2.0 arrays (Affymetrix, Santa Clara, CA, USA) according to Affymetrix’s protocol at the Centre for Genomic Sciences of the University of Hong Kong, as previously reported by our group [51]. Briefly, approximately 8 × 10^5^ KYSE150 cells were seeded in 75 cm^3^ flasks and were allowed to grow for 24 h at 37 °C with 5% CO_2_. After 24 h of incubation, cells were treated with 9.5 μg/mL 91b1 or DMSO (0.05%, Sigma-Aldrich, St Louis, MO, USA) as the blank control, and then were incubated at 37 °C with 5% CO_2_ for 48 h. Total RNA was extracted using a RNeasy Mini Kit (Qiagen, Frankfurt, Germany). The RNA integrity was measured by the ratio of 28S/18S ribosomal RNA by Agilent 2100 Bioanalyzer (USA). cDNA was synthesized from 1 μg of total RNA by reverse transcription kit (Invitrogen, Waltham, MA, USA). Biotin-labeled cRNA was produced by an in vitro transcription kit (Invitrogen, Waltham, MA, USA), and was then purified by RNeasy mini columns (Qiagen, Frankfurt, Germany). About 15 μg denatured cRNA was hybridized to each Human Genome U133 Plus 2.0 array (Affymetrix), and then was stained by a streptavidin phycoerythrin conjugate. The signals were detected by a GeneArray scanner (Agilent, Santa Clara, CA, USA) and were analyzed by Agilent Genespring GX and Affymetrix GeneChip Operating Software. The signals of the differentially expressed genes of the treated samples were compared with the corresponding blank controls. The threshold levels of the corresponding up- or downregulated genes with ≥2 fold changes were included for further analysis.

### 4.10. Quantitative Real-Time PCR

The GoTaq qPCR system (Promega, Madison, WI, USA) was used to analyze the relative mRNA expression of target genes by quantitative real-time PCR. Total RNA was extracted from scratched cells by RNeasy Mini Kit (Qiagen, Venlo, The Netherlands) and reverse-transcribed into cDNA by the GoScript^TM^ Reverse Transcription System (Promega, Fitchburg, WI, USA), according to the manufacturer’s instructions. qPCR reactions were carried out by the PikoReal Real-Time PCR System (Thermo Scientific, Waltham, MA, USA). The Cq (cycle of quantification) of each sample was determined and recorded by the program PikoReal Software 2.0 (Thermo Scientific, Waltham, MA, USA).

For all the qPCR reactions, the relative expressions of target genes in different samples were calculated and compared by using the 2^−ΔΔCt^ method. The expression level of target genes was normalized by the reference gene *β-actin*. Primers for *Lumican:* 5′-CTTCAATCAGATAGCCAGACTGC-3′ (forward) and 5′-AGCCAGTTCGTTGTGAGATAAAC-3′ (reverse). Primers for *β-actin*: 5’-GTGGGGCGCCCCAGGCACCA-3′(forward) and 5’-CTCCTTAATGTCACGCACGATTTC-3’ (reverse).

The calculation of 2^−ΔΔCt^ method was as follows [52]:ΔCq of target gene = Cq of target gene − Cq of reference gene
ΔΔCq of target gene = ΔCq of the target gene in treated group − ΔCq of the target gene in control group

Therefore, the fold change of gene expression level = 2^−(ΔΔCq of target gene)^

The expression level was regarded as overexpression if the fold change of the gene expression level ratio ($\frac{\mathrm{Target}\;\mathrm{gene}\;(\mathrm{tumor})/\mathrm{Reference}\;\mathrm{gene}\;\left(\mathrm{tumor}\right)}{\mathrm{Target}\;\mathrm{gene}\;\left(\mathrm{non}-\mathrm{tumor}\right)/\mathrm{Reference}\;\mathrm{gene}\;\left(\mathrm{non}-\mathrm{tumor}\right)}$) was larger than 1.2; a ratio between 0.8 and 1.2 was considered as no significant change, while a ratio smaller than 0.8 was considered as underexpression of the target gene [53].

### 4.11. Wound-Healing Assay

A wound-healing assay was performed to evaluate cell migration and growth. Approximately 1 × 10^6^ KYSE150 cells were cultured in a 6-well plate at 37 °C with 5% CO_2_ overnight to let the cells adhere and grow to reach about 70~80% confluent monolayers. On the second day, the monolayer was gently scratched with a new 1 mL pipette tip across the center of the well to generate a wound area without changing the medium. After scratching, the well was gently washed twice with warm PBS buffer to remove detached cells, and the well was replenished with fresh medium or different concentrations of tested compounds (compound 91b or recLumican). The cells were incubated at 37 °C with 5% CO_2_ again and observed by microscope (Olympus CKX41, Tokyo, Japan) at different time points (0, 6, 12, 24, or 48 h after scratching, depending on different cell types) for photography.

### 4.12. Transwell Invasion Assay

Cancer cell invasion was evaluated by matrigel-coated membrane (8 μm pore size, BD Biocoat, Corning, San Jose, CA, USA) chambers in a 24-well plate. KYSE150 cells were investigated in this test. The lower chamber was filled with RPMI-1640 medium containing 10% FBS with purified recLumican, Beijing, at concentrations of 0 ng/mL, 50 ng/mL, 250 ng/mL, and 500 ng/mL. Approximately 5000 cancer cells were cultured in 200 μL serum-free RPMI-1640 culture medium in the upper chamber. At the same time, the same number of cells were cultured in an uncoated membrane (8 μm pore size) chamber as a control. After 24 h of incubation at 37 °C with 5% CO_2_, the uninvaded cells on the upper chamber were scraped off with a cotton swab, while the transmembrane cells that migrated to the opposite side of the membrane were fixed in 100% methanol for 10 min followed by staining with 0.5% crystal violet solution after washing twice with PBS. The transmembrane cells were counted under a microscope (Olympus CKX41, Japan) in four random fields at a magnification of 40 times. The invasion percentage was calculated by: invasion% = $\frac{\mathrm{number}\;\mathrm{of}\;\mathrm{cells}\;\mathrm{invading}\;\mathrm{through}\;\mathrm{matrigel}-\mathrm{coated}\;\mathrm{membrane}}{\mathrm{number}\;\mathrm{of}\;\mathrm{cells}\;\mathrm{invading}\;\mathrm{through}\;\mathrm{uncoated}\;\mathrm{membrane}}$ × 100 to determine the cell invasion.

### 4.13. In Vivo Study in Nude Mice Xenograft Model

An in vivo nude mice xenograft model was used to evaluate the anticancer activity of the compound 91b1 in animals. For the preparation of cancer cell xenografts, approximately 1 × 10^6^ trypsinized cells suspended in HBSS (Life Technologies, Waltham, MA, USA) were implanted subcutaneously into the mid-dorsal region of each athymic nude mouse (BALB/c-nu/nu, female, 4 weeks old, purchased from Beijing Charles River Laboratories. The animal approval code was 440072000011798 and the certificate number was SCXK (Beijing) 2012-0001). Tumors were allowed to grow without treatment for 10 days. When the tumors became palpable (approximately 150 mm^3^ in volume, calculated by the formula following [54]), each test agent (0.2 mL in volume, 50 mg/kg/day for the compound 91b1, 1 mg/kg/day for doxorubicin, or vehicle control) was injected into each mouse via the intraperitoneal (i.p.) route. Compound 91b1 was dissolved into 6% PEG and then physiological saline was used to prepare the stock solution to test its anticancer action. Doxorubicin was dissolved in 6% PEG physiological saline as the positive control, and 6% PEG was dissolved in physiological saline as the vehicle control. Each agent was given to each mouse from the tested groups as treatment, n = 5.

Tumor dimensions were assessed every other day with calipers, and tumor volumes were estimated using two-dimensional measurements of length and width and calculated with the formula [l × w^2^] × 0.52 (where l is length and w is width), as previously described [54]. Photos were taken every five days. After 25 days of treatment, all animals were sacrificed by CO_2_ inhalation and then dissected to collect the subcutaneous xenografts.

### 4.14. Statistical Analysis

Two-tailed *t*-test was used to determine the statistical significance of the differences observed between groups. Statistical analyses were conducted by the statistics program GraphPad Prism 5 (GraphPad Software Inc., San Diego, CA, USA) or software Excel. A P-value of <0.05 was considered statistically significant and marked with a*, and a *p*-value of <0.01 was considered remarkably statistically significant and marked with a**.

The comparative ΔΔCt method was applied for relative quantification in qPCR analysis [52].

A violin plot, heatmap, and scatter plot of different expressed genes were analyzed in the R platform (V4.1.0) and visualized by ggplot 2 (V3.36).

## 5. Conclusions

The novel quinoline compound 91b1 demonstrated strong anticancer effects, both in vitro and in vivo. Compound 91b1 suppressed cell proliferation, modulated the cell cycle, and downregulated *Lumican* mRNA expression. The predicted target of compound 91b1 *Lumican* was found to be overexpressed in many kinds of cancer cells, and induced cancer cell migration and invasion. It is hypothesized that compound 91b1 inhibits cancer cell progression by downregulating *Lumican* expression. The above results suggest the great potential of quinoline compound 91b1 to be developed as a novel anticancer drug, and they indicate that *Lumican* could be developed as a new therapeutic target in cancer treatment.

## Figures and Tables

**Figure 1 ijms-23-13181-f001:**
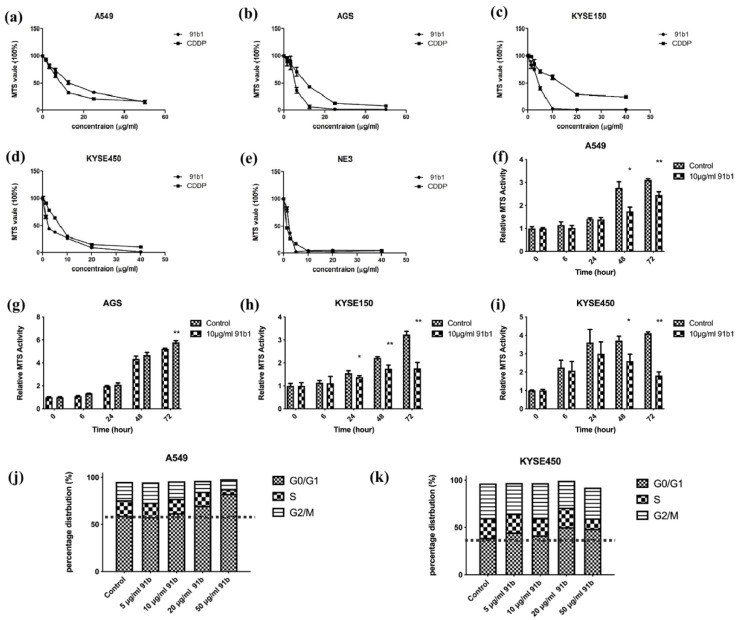
Compound 91b1 showed anticancer effect in vitro. Compound 91b1 and CDDP both showed dose-dependent cytotoxicity for the (**a**) A549 cell line; (**b**) AGS cell line; (**c**) KYSE150 cell line; (**d**) KYSE450 cell line; and nontumor cell line (**e**) NE3 cell lines. CDDP was used as the positive control. N = 4. Compound 91b1 inhibited cell proliferation of cancer cells of (**f**) A549 cells; (**g**) AGS cells; (**h**) KYSE150 cells; and (**i**) KYSE450 cells cultured with 10 μg/mL of compound 91b1. N = 4. Vehicle control: 0.1% DMSO (dimethyl sulfoxide). CDDP: cisplatin. * *p* < 0.05; ** *p* < 0.01; (**j**): cell-cycle distribution of A549 cells and (**k**): KYSE450 cells after treatment of increasing concentrations of compound 91b1. 0.1% DMSO was applied as vehicle control, N = 3. * *p* < 0.05; ** *p* < 0.01.

**Figure 2 ijms-23-13181-f002:**
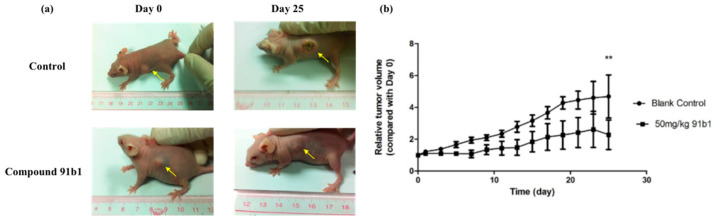
Compound 91b1 inhibited tumor growth in vivo. (**a**) Images of one animal from the vehicle control group and compound 91b1-treated group in the KYSE150 xenograft test on the first day and the 25th day; (**b**) relative tumor volume changes of subcutaneous KYSE150 xenografts of the vehicle control group and compound 91b1-treated group (50 mg/kg/day compound 91b1) after 25 days; 6% PEG saline was applied as the vehicle control. N = 5. ** *p*-value < 0.05.

**Figure 3 ijms-23-13181-f003:**
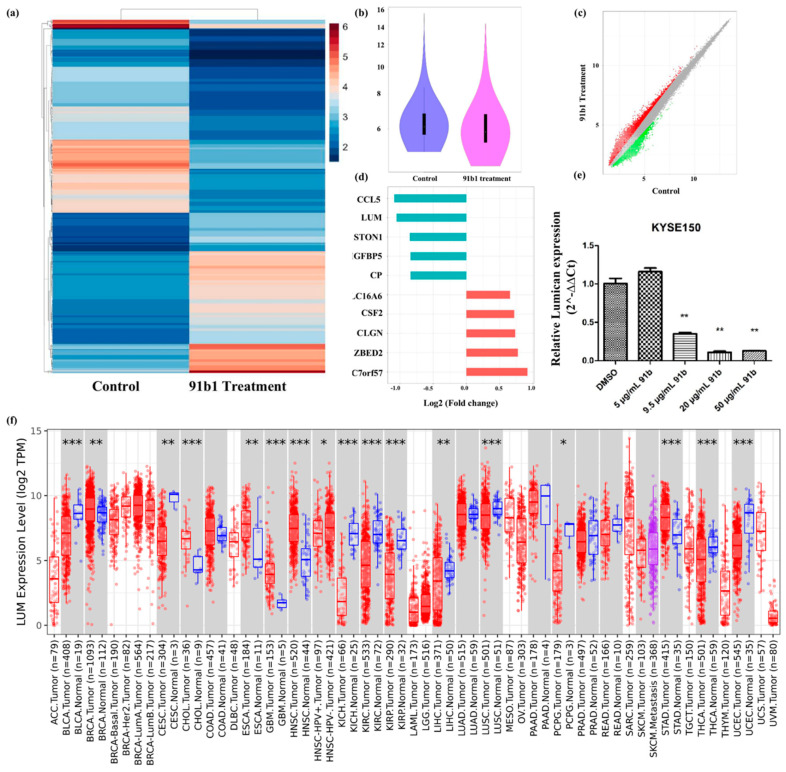
*Lumican* plays a critical role in several kinds of cancers and can be downregulated by compound 91b1. (**a**) Heatmap of differentially expressed genes in KYSE150 cells treated with compound 91b1 versus blank medium control, the color indicates the relative expression of genes according to color bar; (**b**) violin plot of microarray data of KYSE150 cells treated with compound 91b1 for 48 h versus blank medium control; (**c**) scatter plot of differently expressed genes in KYSE150 cells treated with compound 91b1 versus blank medium control, red dots indicate up-regulated genes, and green dots indicate down-regulated genes; (**d**) most genes regulated by compound 91b1 identified by microarray analysis; (**e**) relative *Lumican* expression level after 48 h of treatment with different concentrations of compound 91b1 (5 μg/mL, 9.5 μg/mL, 20 μg/mL, and 50 μg/mL) and vehicle (0.1% DMSO) in KYSE150. The relative *Lumican* expression level was determined by comparing with cells treated with the vehicle after normalization with the expression of β-actin using qPCR. ** *p* < 0.01; (**f**) pancancer analysis of *Lumican* by TIMER2.0, up-regulated or down-regulated genes in the tumors were compared to normal tissues for each cancer type, red dots indicate tumor tissue, bule dots indicate normal tissue, purple dots indicate to metastasis tumor tissue, * *p* < 0.05; ** *p* < 0.01; *** *p* < 0.001 [33].

**Figure 4 ijms-23-13181-f004:**
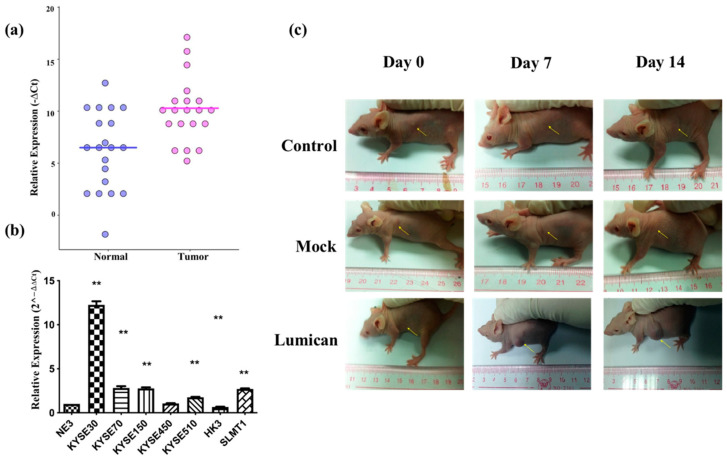
*Lumican* is usually overexpressed in tumor tissue and several kinds of cancer cell lines, and promotes tumorigenesis. (**a**) Relative *Lumican* expression level in tumor tissue and adjacent normal tissue isolated from cancer patients. β-actin was applied as the reference gene to normalize the *Lumican* expression. N = 20. ** *p* < 0.01; (**b**) relative *Lumican* expression levels in seven cancer cell lines and the nontumor cell line (NE3). The relative *Lumican* expression level was determined by comparison with NE3, after being normalized with the expression of β-actin. ** *p* < 0.01; (**c**) images of subcutaneous tumor formation in the nude mice with the injection of NIH 3T3 parental cells, Mock vector, or *Lumican* gene transfected NIH 3T3 cells on day 0, day 7, and day 14, respectively, after injection.

**Figure 5 ijms-23-13181-f005:**
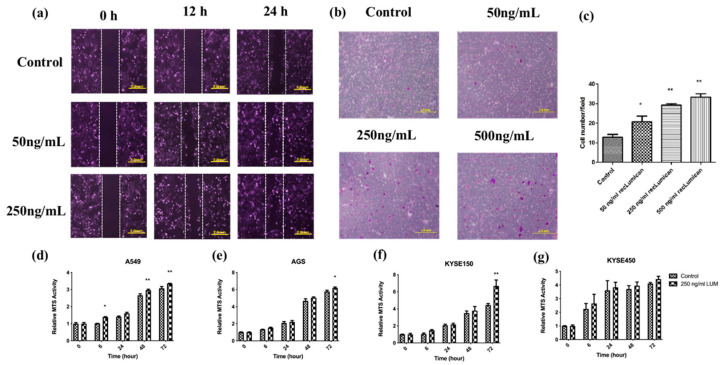
*Lumican*-induced cancer cell migration, invasion, and proliferation. (**a**) Images of wound-healing assay with recLumican protein treatment (50 ng/mL or 250 ng/mL) on KYSE150 cells at 0 h, 12 h, and 24 h. Exposure time: 12.5 ms. Original magnification: 10×. Scale: 2 mm. (**b**) Cell invasion assay with the transwell matrigel chamber and KYSE150 cells cocultured with different concentrations of purified human recLumican. Original magnification: 40×. (**c**) Average invaded cell numbers of KYSE150 cocultured with different concentrations of purified human recLumican (0, 50, 250, 500 ng/mL recLumican). The invaded cells were counted under a microscope in four random fields at the original magnification of 40×. * *p* < 0.05; ** *p* < 0.01. Proliferation curves of cancer cells cocultured with or without 250 ng/mL human recLumican of (**d**) A549 cells; (**e**) AGS cells; (**f**) KYSE150 cells; (**g**) KYSE450 cells. N = 3. Fresh culture medium was applied as the blank control. * *p* < 0.05; ** *p* < 0.01.

**Figure 6 ijms-23-13181-f006:**
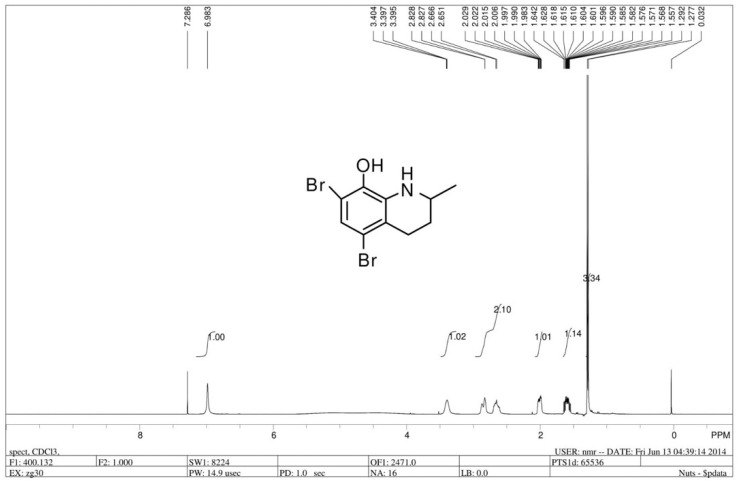
^1^H-NMR spectrum of compound 91b1.

**Table 1 ijms-23-13181-t001:** MTS_50_ value (μg/mL) of compound 91b1 and CDDP for four cancer cell lines and one nontumor cell line (NE3). Results were calculated by GraphPad nonlinear regression analysis from four parallel experiments.

Cell Lines	MTS_50_ Value (μg/mL)	
	91b1	CDDP
A549	15.38	6.23
AGS	4.28	13.00
KYSE150	4.17	13.2
KYSE450	1.83	6.83
NE3	2.17	1.17

CDDP: cisplatin. N = 4. Vehicle control: 0.1% DMSO (dimethyl sulfoxide).

## Data Availability

Not applicable.

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
