# Peer review of "The Anticancer Effect of a Novel Quinoline Derivative 91b1 through Downregulation of Lumican"

_ijms, 2022, doi:10.3390/ijms232113181_

Round 1
Reviewer 1 Report
In the manuscript by Zhou et al., the authors investigated the effect of a novel compound 91b1, on cell proliferation and cancer. They identified several targets of 91b1 and characterized Lumican as a potential target through which 91b1 might regulate cell proliferation. Specifically, the authors discover that 91b1 suppresses proliferation by inhibiting Lumican. Overall the findings are interesting, and before accepting for publication, the following comments need to be addressed.
Comments:
- Fig 3 e and f did not have legends. Please add figure legends for this.
- What cells are used for the experiments in figure 5a that need to be added to the figure legend?
- In the sentence " Collectively, there were 31,520 up-regulated genes, and 13,180 down-regulated genes were identified in total," Are the authors referring to genes or mRNA?
- "The function of the top downregulated gene Ccl5 was investigated by Dr. Dessy Chan" please cite the paper.
Reviewer 2 Report
The article's authors described the down-regulation of Lumican by using the quinoline derivative obtained in the authors' group. This phenomenon is described as to be an anticancer effect.
In the introduction, the authors describe the health threat of cancer as a worldwide issue. Cancer studies aim to identify the proper oncogenes as potential targets for novel therapies. The authors also indicate that the main goal is low toxicity and high efficacy of anticancer drugs as one of the potential classes of compounds that could be an excellent example of anticancer drugs the quinoline derivatives are indicated. Several examples with proper citations are mentioned.
Another critical issue in cancer drug discovery is the mechanism of the anti-tumour agent. In the case of quinolines, it is well studied and investigated topic. As the model target authors used Lumican gene because LUMICAN levels and Lumican expression is found to be correlated with breast and lung cancer.
The article is clear, and the study's primary goal is well described. Nevertheless, there are some issues that I would like to mention so that the authors can reply.
I would like the authors to explain why they chose this specific derivative. Why has only one compound been studied in this article? If some other compounds were synthesized, maybe a range of modified quinoline derivatives would show similar or better properties as an anticancer agent.
Regarding cytotoxicity, the authors compared it with the well-known drug CDDP (cisplatin). It is a good example of comparing new possible drugs. I would suggest that it would be valuable for this study to include some other quinoline derivative or even unmodified quinoline. It would be a good indication of how the implemented modification in the compound structure influenced its anti-tumour properties and overall effect on the tested cell line.
Again back to compound 91b1, the authors mention that it was synthesized by Dr Penny Chan, who is not one of the authors of the reviewed articles. If that was the authors' intention, I would expect some acknowledgement for Dr Chan at the end of the article because I understand that Dr Chan contributed her work in different articles and is not interested in being a coauthor in this article.
Subsection 4.2 indicate that the compound was examined with MS, but there is no mention of it in the actual text.
The authors should change something in figure 1, because the control and experimental curves are almost indistinguishable on all graphs a-i. The same issue is with figure 5. d-g.
The authors should be more considerate using terms like "mechanism at molecular level". To my understanding, this study showed an effect of using a particular chemical on cancer tissue and showed anticancer potential and down-regulation of LUMICAN. The actual mechanism of how 91b1 interacts with tumour tissue is unknown and needs further investigation. There is no explanation the text of the interaction of 91b1 at the molecular level. I ask the authors to explain how 'this study investigate the related mechanism at molecular level'.
I also found some minor editorial mistakes in the text, but there may be more:
Line 48-49 & 49-50: repetition of the same sentence
Line 320: emantiomer (should be enentiomer)
I believe that the English language should be revised in the article.
The article should be revised once more after all answers and changes are made by the authors.
Reviewer 3 Report
The manuscript (ijms-1958215) presented by Zhou et al. describes the interesting topic of the anti-cancer activity of the novel quinoline derivative (compound 91b1) and the role of Lumican in this process. Although quinoline derivatives constitute a broad topic of consideration in the context of anti-cancer therapy, this manuscript cannot be accepted in its present form. The major reasons are:
- A significant number of references refer to publications older than 10 years.
- In the publication cited by the authors (Ref. nr 30) on the synthesis of compounds, the reviewer did not find any information about compound 91b1 which is discussed in this manuscript. No description of the synthesis or structure of the compound was found in the presented manuscript.
- According to the reviewer, incorrect interpretation of the cell cycle analysis and disturbed control sample in which the number of cells in the G2/M phase is 60%. In proper control of the cell cycle, the number of cells in the G2/M phase is approx. 15-20%.
- A discussion that is a description of the results, not a discussion with the current research results of other scientists.
- No explanation as to why doxorubicin and cisplatin were used as positive controls.
- The manuscript needs a lot of editorial corrections as well as extensive editing of English language and style required.
Regards
Round 2
Reviewer 2 Report
I want to thank the authors of this manuscript for their work and every correction they made in their article.
All answers provided by the authors and manuscript changes were correct and improved the article's quality.
To my knowledge, the present shape of the manuscript should be forwarded to publishing after correcting every misspelling and a minor error in the text.
Reviewer 3 Report
Reviewer accept the manuscript after author`s revisions.
